# Efficient Metric for Distinguishing Memorization from Generalization in Large Language Models

## Abstract

This work proposes a computationally inexpensive method to measure memorization of training data in LLMs (Large Language Models) while accounting for generalization. Prior approaches such as counterfactual memorization Zhang et al. (2023), have been computationally expensive, and therefore only been studied in limited settings. However, our new metric, Prior-Aware memorization, does not require training any new models, and can thus be directly applied to existing LLMs trained on large amounts of data. We evaluate our metric on two pre-trained models, Llama and OPT, trained on the Common Crawl and The Pile, respectively. We discover that for the largest models, $55 - 90\%$ of the sequences that would be classified as "memorized" in earlier models are, in fact, generalizable sequences.

## 1 Introduction

Training data leakage from Large Language Models (LLMs) has been a concern for many reasons. Two important concerns include a) copyright and licensing violations, which have been the subject of several lawsuits and prior literature Chang et al. (2023); Kadrey v. Meta Platforms, Inc.; Karamolegkou et al. (2023); The New York Times Company v. Microsoft Corporation; Tremblay v. OpenAI, Inc., and b) leakage of sensitive data, such as Personally Identifiable Information (PII) Carlini et al. (2021); Mozes et al. (2023). Starting with Carlini et al. (2021), several studies have attempted to quantify LLM training data leakage, often using various metrics to benchmark the extraction of training data in different training and inference scenarios Biderman et al. (2024); Carlini et al. (2022); Karamolegkou et al. (2023); Yu et al. (2023).

Prior approaches to quantifying memorization in LLMs often overlook the models' capacity to generalize, conflating genuine memorization with the generation of statistically common sequences Carlini et al. (2022); Yu et al. (2023). For example, a prompt like "The murder was committed by" may yield "John Doe" with high probability not because the model memorized this sequence from training, but because "John Doe" is a common placeholder name. Recent work confirms that LLMs can reproduce text verbatim by generalizing from related patterns rather than recalling specific examples Liu et al. (2025). Thus, statistically likely sequences should not be classified as memorized.

A current, robust approach to account for such generalization is *Counterfactual Memorization* Zhang et al. (2023). Counterfactual memorization is traditionally measured by computing the difference between the likelihood of leaking a training sequence from a model that was trained with versus without a given sequence. If a given sequence has a high probability of leakage in both models, then it is probably not counterfactually memorized. Naturally, an explicit computation of this difference may be very expensive, as it requires us to train several "counterfactual" or "baseline" models for every training sequence. These measurements also often necessitate tightly controlled settings, making them very impractical to reproduce on production language models Zhang et al. (2024). As a result, prior studies on counterfactual memorization have been limited in scale Ghosh et al.; Zhang et al. (2023). This motivates the need for a cheaper method for filtering out "popular" sequences.

To address this limitation, we introduce a new criterion for filtering out statistically likely outputs (suffixes) by assessing whether they are strongly associated *only* with their specific training prompts (prefixes). We term this phenomenon *Prior-Aware memorization* (PA memorization). This framework distinguishes genuine memorization from statistical likelihood by testing whether a sequence

retains high generation probability even when the model is prompted with randomly sampled text from the training data. If a suffix appears with high confidence across many unrelated prefixes, its likelihood arises from generalization—reflecting statistical commonality—rather than from memorization of a specific prefix–suffix pair.

A key advantage of PA memorization is that it requires no additional model training, making it substantially more computationally efficient than counterfactual memorization. Furthermore, to provide evidence that PA memorization effectively filters out statistically likely sequences, we present an empirical comparison with Counterfactual Memorization Zhang et al. (2023).

We evaluate our methodology on text from the training corpus of two pre-trained models: Llama, and OPT. Our target sequences were either a) randomly sampled long sequences (to simulate copyright-infringement scenarios), or b) Named Entities (*e.g.,*names of individuals, places, etc.) to simulate the risk of leaking Personally Identifying Information (PII). We find that using the largest OPT and LLama models, $55 - 90\%$ of the sequences that would earlier be labeled as memorized are in fact statistically common sequences. More surprisingly, we note that a similar result occurs even with the SATML training data extraction challenge dataset Yu et al. (2023), where around $40\%$ of sequences are "common" in nature. This is despite the fact that each sequence in the challenge dataset occurs only once in the entire training data. Our findings highlight the significance of looking beyond the low-frequency of exact copies in the training data when labeling a sequence as memorized, and urge us to rethink some of our previous notions about memorization in LLMs.

The remainder of this paper is organized as follows: in Section 2, we outline the current research gaps to motivate the need for our work. Section 3 includes the definitions of the metrics we use for our experimental results. We present our findings in Section 4 and highlight any novel conclusions. Section 5 outlines related literature, and we conclude this work in Section 6.

## 2 GENERALIZATION VS MEMORIZATION: ISSUES WITH LEAKAGE MEASURES

Most prior work considers training data that can be produced verbatim with high probability to be "memorized" Biderman et al. (2024); Carlini et al. (2021; 2022); Yu et al. (2023). However, as noted above, such methods may misclassify popular sequences as memorized. To make this limitation precise, consider a prefix $p$ and suffix $s$, with $p\|s \in D$ indicating that their concatenation occurs in the dataset $D$. Existing metrics typically label such a sequence as memorized if the conditional probability $P(s \mid p)$ is large Carlini et al. (2022); Yu et al. (2023), which we refer henceforth to *extractable memorization*. Extrctable memorization is reasonable: under maximum likelihood training, the model's objective is to maximize $\prod_{(p,s)\in D} P(s \mid p)$, so if a model has genuinely memorized a given $p\|s$, then $P(s \mid p)$ should indeed be high.

However, the converse is not necessarily true—i.e., a high conditional probability does not necessarily imply memorization. This is because $P(s \mid p)$ can be high for two distinct reasons, which we formally decompose using Bayes' rule:

$$P(s \mid p) = \frac{P(p \mid s) \cdot P(s)}{P(p)} \tag{1}$$

Indeed, $P(s \mid p)$ can be high if:

1. If $P(s)$ is very large, suggesting that $s$ may be statistically popular (generic). For instance, in the sequence if $p$ = "The murder was committed by" and $s$ = "John Doe", $P(s)$ may be large because $s$ is a popular occurrence in the dataset.

2. If the relative belief ratio $\frac{P(p|s)}{P(p)}$ is large. This would happen if $P(p)$ is small (*e.g.,*the prefix is a rare prefix in the dataset) relative to $P(p \mid s)$, suggesting that the suffix is a strong indicator of the prefix, as we would expect for memorization.

There have been alternative definitions of memorization, such that of Schwarzschild et al. (2024), which classifies a sequence as memorized if it can be generated with a compressed prompt. Similarly, concurrent work by Morris et al. (2025) also defines a compression-based memorization metric

using Kolmogorov complexity (which is not computable in general). However, it is unclear whether these alternative approaches can distinguish sequences $p\|s$ with statistically popular targets $s$ from their truly memorized counterparts.

Recently, a stronger definition of memorization has been proposed, called *Counterfactual Memorization* Zhang et al. (2023). This definition measures how models trained with and without the target sequence would perform on the sequence. A counterfactually memorized sequence is then categorized as a sequence that would not perform well on a model that was not explicitly trained on said sequence. If a sequence performs well on both models, it is said to be *generalizable*, since it can be produced by a model that was not trained on the sequence.

Failing to account for generalization capabilities may greatly overestimate memorization, as indicated by a recent study Liu et al. (2025). However, accurately computing Counterfactual Memorization can also be extremely expensive, as the model must be trained at least twice for every labeled sequence–once with the sequence, and, second, on the same training data absent the specific sequence. Naturally, for models with billions of parameters trained on many terabytes of text, this can quickly become infeasible. One may consider cheaper estimates to this calculation, such as training approximate baselines with a different data distribution; however, it is fundamentally difficult to make these estimates accurate Zhang et al. (2024).

All told, there is a clear need for an inexpensive, but accurate, method for identifying whether a particular training sequence is Counterfactually Memorized.

## 3 Our Metric: Prior-Aware (PA) Memorization

We next propose and describe our approach to quantifying *Prior-Aware* [ PA ] memorization, starting with a definition of this metric and proceeding to discuss how it can be feasibly computed. We empirically test our metric's ability to filter out generic sequences by measuring its correlation Counterfactual Memorization from the literature Zhang et al. (2023).

Throughout this section, we presuppose a language model $M$, defined as follows.

**Definition 1.** *The language model $M$ maps an input token sequence $p = (t_1, t_2, \ldots, t_j)$ to a distribution for the next token $t_{j+1}$:*

$$M : p \mapsto P(\cdot \mid p).$$

We assume that the model $M$ is trained on a dataset $D$, consisting of sequences of prefix + suffix pairs $p\|s$ (where $\|$ is the concatenation operator). We also assume that text generation proceeds auto-regressively from a starting sequence $y$, and each time generating (and incorporating into $y$) a subsequent token. We will use the notation $P(x|y; M)$ to denote the probability of generating sequence $x$ from such an application of $M$ to an initial sequence $y$, and $P(x; M)$ will denote the total probability of seeing $x$ from any starting sequence. Where the choice of model is clear, we may omit it for brevity, and write simply $P(x|y)$ or $P(x)$.

**Definition 2** (Prior-Aware memorization). *For tunable thresholds $m, n \geq 0$ we say that a sequence $p\|s \in D$ is* Prior-Aware memorized *by a model $M$ if $P(s \mid p; M) > m$, and $\frac{P(s|p;M)}{P(s;M)} > n$.*

**Explanation.** The definition imposes two requirements: First, that $P(s \mid p; M) > n$ (see Section 3.1), which ensures that the suffix $s$ has a high probability of being generated verbatim by model $M$ when prompted with $p$. This has been adopted from prior work Carlini et al. (2022). Second, that the relative beleif ratio, $\frac{P(s|p;M)}{P(s;M)} > m$ (see Section 3.2), implying that $s$ is a strong indicator of $p$. A large ratio ($m >> 1$) suggests that the presence of $s$ predicts the presence of $p$, or, in other words, $s$ was less likely to be generally produced by another sequence.

To compute this metric for any given pair $p\|s \in D$, we need to compute $P(s \mid p; M)$ and $P(s; M)$. We explain how to do this in the next two subsections.

### 3.1 $P(s \mid p; M)$: verbatim generation of $s$ given $p$

For notational convenience, we write $t_{1:j}$ to denote the sequence of tokens $(t_1, t_2, \ldots, t_j)$. Using this notation, the probability of producing a sequence of $k$ tokens $s = t_{j+1:j+k}$ (suffix) when the

model is prompted with a $j$-token prefix $p = t_{1:j}$ is:

$$P(s \mid p) = \prod_{i=j+1}^{j+k} P(t_i \mid t_{1:i-1}) \tag{2}$$

Equation 2 is a closed form for the probability of leaking our target sequence verbatim from a given prefix. In practice, this metric can be computed from a single forward-pass per token, and we can thus efficiently utilize this to compute whether $P(s \mid p) > n$ for a given $p\|s$.

## 3.2 $P(s; M)$: GENERALITY OF $s$

The second component of our definition is $\frac{P(s|p)}{P(s)}$. To compute the denominator, we would like to calculate the total probability over all prefixes $V^*$ in the model's vocabulary $V$.

$$P(s) \hat{=} v_s = \sum_{p_i \sim V^*} P(s \mid p_i) \cdot P(p_i) \tag{3}$$

However, this calculation is computationally prohibitive. Instead, use an unbiased estimator $\hat{v}_s$ (inspired by Monte-Carlo Integration Binder et al. (1992)) that converges to the true probability $P(s)$ as the number samples $c$ grows to infinity.

**Definition 3.** *Given samples $q_0, q_1, \ldots q_{c-1}$ chosen independently and identically according to the distribution of prefixes $p_i \sim V^*$, our total probability estimator is given by*

$$\hat{v}_s = \frac{1}{c} \sum_{i=0}^{c} P(s \mid q_i) \tag{4}$$

Note this estimator requires samples $q_i$ to match the distribution of prefixes $p_i \sim V^*$, and it is thus computed for a specific model $M$; for brevity, we omit $M$ from the notation where it can be straightforwardly inferred. The next two theorems show that $\hat{v}_s$ is unbiased.

**Theorem 1.** $E[\hat{v}_s] = v_s$.

*Proof.* By linearity of expectation, $E[\hat{v}_s] = \frac{1}{c} \sum_{i=1}^{c} E[P(s \mid q_i)]$.

Since $q_i$ are chosen i.i.d, we have that

$$E[P(s \mid q_i)] = E[P(s \mid q_1)].$$

Thus,

$$\begin{aligned} E[\hat{v}_s] &= \frac{1}{c} \cdot (c \cdot E[P(s \mid q_1)]) \\ &= E[P(s \mid q_1)] \\ &= E[P(s \mid p_1)] \\ &= v_s \end{aligned}$$

The transition from conditioning on $q_1$ to $p_1$ follows because the samples $q_i$ are chosen according to the same distribution as $p_i$. $\square$

Theorem 2 shows that $Var[\hat{v}_s] \to 0$ as $c \to \infty$. This means that over a large number of prefixes sampled from the dataset, the error of the estimator tends to 0 for Theorem 2. We state the theorem below, but relegate its proof to Appendix A due to space constraints.

**Theorem 2.** $Var[\hat{v}_s] = \frac{1}{c} \cdot Var[P(s \mid p_i)] \le \frac{1}{4c}$

## 3.3 EMPRICIAL CORRELATION WITH COUNTERFACTUAL MEMORIZATION

In this section, we empirically show that PA memorization does indeed filter out $p\|s$ where $s$ is statistically popular by correlating it with counterfactual memorization Zhang et al. (2023), another metric that can measure statistically likely sequences. To do so, we train several models in a controlled setting, and measure both counterfactual memorization and $\frac{P(s|p)}{\hat{v}_s}$. We then show that these are positively correlated.

| Exact Copy | Near-Dup |
|---|---|
| Quantum doughnuts might not exist, but theoretical bakers remain hopeful. | majestic*um* Nant*onuts* might Conradavery 258 texted *theoretical* imperialistmlicks Shim. |

Table 1: A sequence and one possible $20\%$ near-duplicate. Matching tokens are highlighted

### 3.3.1 Counterfactual Memorization

Below, we restate the definition of Counterfactual memorization from Zhang et al. (2023).

**Definition 4** (Counterfactual Memorization). *Given a training algorithm $A$ that maps a training dataset $D$ to a trained model $f$, and a measure $L(f, x)$ of the accuracy of $f$ on a specific example $x$, the counterfactual memorization of a training example $x$ in $D$ is given by*

$$\text{mem}(x) \triangleq \underset{S \subset D, x \in S}{E}[L(A(S), x)] - \underset{S\prime \subset D, x \notin S\prime}{E}[L(A(S\prime), x)] \tag{5}$$

where $S$ and $S\prime$ are subsets of training examples sampled from $D$. The expectation is taken with respect to the random sampling of $S$ and $S\prime$, as well as the randomness in the training algorithm $A$. Generally speaking, the first term measures the expected accuracy over models that contain the target sequence $x$, and the second measures the same over models that do not contain $x$.

### 3.3.2 Model Training.

We train 124M parameter GPT-2 models with 1000 Wikitext documents in two settings, as described below. We repeat the experiments in each setting several times with different seeds and target sequences, training over 350 different models.

**Target Model Setting:** In this setting, we inject "near-duplicate" sequences into the dataset, a strategy commonly used to study memorization versus generalization in prior work Liu et al. (2025); Zhang et al. (2023). For our purposes, near-duplicates are defined as sequences sharing $20\%$ token overlap, as illustrated in Table 1. Unlike earlier studies, which often used duplicates with much higher overlap, we intentionally adopt a lower threshold to be conservative. This choice is motivated by recent findings that removing high-overlap sequences improves model performance Lee et al. (2022). Thus, injecting highly overlapping sequences may not realistically reflect modern training regimes, where such examples are typically removed Touvron et al. (2023); Zhang et al. (2022). Our goal is to demonstrate that even with relatively low overlap, models are capable of generalizing to a given sequence without requiring many (or any) exact copies in the training data.

To simulate varying degrees of generality, we vary the ratio of exact copies to near-duplicates of a target sequence injected into the training data. For example, an initial dataset may contain 180 near-duplicates of a target sequence and no exact copies. In the next dataset, 30 of these near-duplicates are replaced with 10 exact copies, yielding 150 near-duplicates and 10 exact matches. The rationale is that as more exact copies are included, the model is more likely to exhibit counterfactual memorization, predicting the target sequence because it has seen it verbatim during training. Conversely, when the dataset consists mostly of near-duplicates, the model must rely on generalization from many similar–but not identical–examples, simulating the case where a sequence is generated with high probability simply because there is a lot of similar data in the training set Liu et al. (2025). By carefully controlling the ratio of exact copies to near-duplicates, we can precisely modulate the extent to which the model relies on memorization versus generalization in order to reproduce the target sequence.

We use the following number of (exact copies, near-duplicates) to create 7 distinct types of datasets: $(0, 180), (10, 150), (20, 120), (30, 90), (40, 60), (50, 30), (60, 0)$. In every case, the total training set size remains fixed at 1000 sequences, with only the composition of exact versus near-duplicate instances varying. We repeat this process for 25 different target sequences, training $25 \times 7 = 175$ different target models.

**Baseline Model Setting:** The counterfactual memorization metric relies on a comparison between the target model and a baseline model. In accordance with the metric's definition of a baseline model, we remove only the exact copies of the target sequence, but keep the near duplicates in the training data. So the baseline model for a target model trained with 150 near-duplicates and 10 exact copies

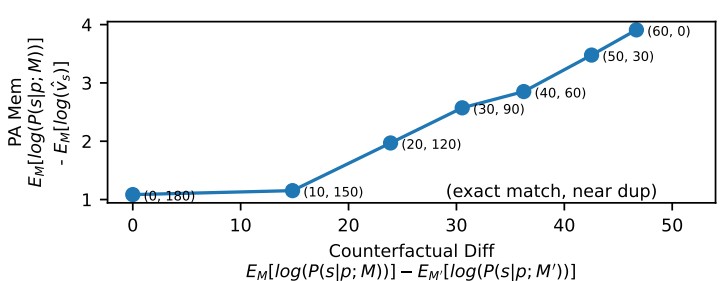

Figure 1: Counterfactual Memorization (x-axis) vs PA Memorization (y-axis). We see that both metrics are positively correlated across different training settings

would have only the 150 near duplicates. This is to simulate the scenario where the exact target data might be removed, but other data that can generalize to the target sequence is still present in the training dataset.

### 3.3.3 METRIC MEASUREMENTS.

We describe how we measure Equations (5) and (2) in our experiments.

**Counterfactual Memorization.** In our setup, we interpret $x$ in (5) as $p\|s$, where $p$ and $s$ are of equal length. To simplify notation, we use $M$ and $M\prime$ to refer to $A(S)$ and $A(S\prime)$ respectively, which we refer to as the target ($M$) and baseline ($M\prime$) models. To measure the accuracy of the model $M(x = p\|s)$, we utilize the measure $\log(P(s \mid p; M))$, following the definition of extractable memorization presented in Carlini et al. (2022). The expectation is taken over all models trained with the same ratio of near-duplicates to exact copies. As in Zhang et al. (2023), we assume these models are uniformly distributed, so the expectation reduces to the average of $\log(P(s \mid p))$ across multiple trained models. Thus, we measure the following:

$$E_M[\log(P(s \mid p; M))] - E_{M\prime}[\log(P(s \mid p; M\prime))] \tag{6}$$

**PA Memorization.** To keep PA memorization comparable with counterfactual memorization, instead of measuring $E_M[\frac{P(s|p;M)}{\hat{v}_{s;M}}]$, we measure its log, like so:

$$E_M[\log(\frac{P(s \mid p; M)}{\hat{v}_s})] = E_M[\log(P(s \mid p; M))] - E_M[\log(\hat{v}_{s;M})] \tag{7}$$

Note that PA memorization only relies on $M$, thus obviating the need for training baseline models $M\prime$ like with counterfactual memorization.

### 3.3.4 RESULTS

**Positive Correlation** Figure 1 shows the counterfactual memorization metric (Equation 6) on the x-axis, and the component of PA memorization that measures the commonality of $s$ (Equation 7) on the y-axis. Each data point represents the average over 25 different models trained. Figure 1 illustrates that as the target sequence becomes less generic due to fewer near-duplicates, the counterfactual memorization metric (Equation (5)) increases correspondingly. As hypothesized, we also observe a strong correlation between PA memorization and counterfactual memorization, suggesting that our metric indeed measures generalization from similar data. One interesting observation is the difference in the scale of the two metrics, a limitation that we discuss in more detail in Section 4.7.

## 4 EVALUATION

### 4.1 EVALUATION MODELS

We evaluate these metrics on various sizes 2 recent generative models with open source datasets: Llama Touvron et al. (2023) and OPT Zhang et al. (2022), which were pre-trained on the Common Crawl (Llama), or The Pile (OPT). We experiment with all available sizes of the models, namely 3B, 7B, and 13B for Llama, and 125M, 350M, 1.3B, 2.7B, 6.7B and 13B for OPT. Unless otherwise stated, the default model size for which results are displayed is OPT 6.7B and Llama 7B.

## 4.2 Target Sequences for Extraction

To compute $P(s)$ for each sequence we test, we prompt the model with 5000 randomly sampled sequences from their respective datasets and measure the likelihood of generating $s$ in each case. We repeat this for 5 trials. Our evaluation is performed in 3 different settings:

**Named Entities.** Following Lukas et al. (2023), we randomly sample $\approx 5000 - 8000$ sequences that contain Named Entities (NEs) from each of the datasets. We do this since NEs can simulate sensitive data that might be targets for extraction since they are typically names of individuals, organizations, places, etc. We must note that this is simply to emulate a realistic use-case; in reality, model developers would repeat the measurements with sequences that they consider sensitive. For consistency, each of our sequences consists of a prefix of 50 tokens followed by a 4-token long Named Entity. While we experiment with longer prefix lengths of upto 400 tokens, unless otherwise stated, the default prefix length is 50 tokens.

**Long Sequences.** We also simulate extraction of longer sequences, which could be useful to measure the risk of leaking copyrighted data. For this, we randomly sample 5000 sequences with 50-token prefixes and 50-token suffixes from each dataset, similar to prior work Biderman et al. (2024); Carlini et al. (2022). Here too, we experiment with longer prefix lengths of upto 400 tokens, but unless otherwise stated, the default prefix length is 50 tokens.

**SATML Challenge.** Lastly, we also perform a smaller-scale, additional analysis on the dataset in Yu et al. (2023), which consists of 1-eidetic sequences (sequences where each $p\|s$ is known to occur only once in The Pile). This dataset was released for the 2023 SATML training data extraction challenge and consists of $15,000$ sequences, each with a prefix of 50 tokens followed by a suffix of 50 tokens. We present results on 1000 of the $15,000$ sequences.

## 4.3 Hyper-parameters

The only parameters that need to be tuned in Definition 2 are $m$ and $n$:

**$m$, threshold for $P(s|p)$:** This parameter allows us to pick which $p\|s$ have a high risk of leakage, and thus can be adjusted based on the risk tolerance. Since $m$ is a threshold for $P(s|p)$, $\frac{1}{m}$ indicates, on average, the number of times a user would need to prompt the model with $p$ to leak the target $s$. We set $m = 0.01$ for 4-token $s$, and $m = 0.0001$ for 50-token $s$. We deliberately pick small values of $m$ to be conservative in our estimation of memorization.

**$n$, threshold for $\frac{P(s|p)}{\hat{v}_s}$:** The second parameter, $n$, allows us to differentiate $p\|s$ where $s$ has a large prior from those that do not. To estimate this, we compute $\frac{P(s|p)}{\hat{v}_s}$ over $s$ that are known to be easy-to-predict for LLMs (examples provided in B). We set $n$ to be the average of $\frac{P(s|p)}{\hat{v}_s}$ over these sequences. Note that since $\frac{P(s|p)}{\hat{v}_s}$ is a model specific quantity, the threshold is recomputed for each model tested.

## 4.4 Effect of Model Size on PA Memorization

In Figures 2 and 4a, we plot both the number of *extractably memorized* (simply high $P(s \mid p)$, as defined by Carlini et al. (2022) and in Section 2) and *PA memorized* sequences as a function of model size. We also plot PA memorized sequences as a proportion of the total extractably memorized sequences. There are two interesting observations to be made:

**Gap in extractable and PA memorized sequences**. Consistent with prior work Carlini et al. (2022); Hayes et al. (2024); Schwarzschild et al. (2024), both extractable and PA memorization increase as the model size increases. However, there is a difference between number of extractably memorized and PA memorized sequences. Notably, this difference is quite large when the target suffixes are 4 token Named Entities, where as few as 10% of extractable memorized samples are PA memorized in the largest models. This suggests that most of the suffixes of extractably memorized sequences are indeed statistically popular rather than truly memorized. As shown in Table 2, many of these $s$ are indeed popular entities such as politicians, celebrities, countries, etc. More surprisingly, we note a similar result in the SATML challenge dataset in Figure 4a, where around 40% of sequences are "common" in nature. This is despite the fact that each sequence in the challenge dataset occurs only once in the entire training data.

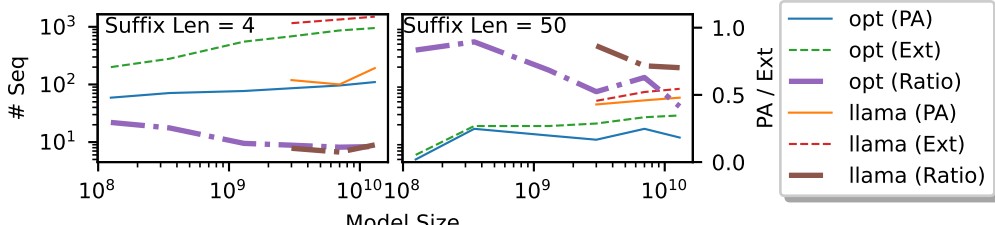

Figure 2: The primary y-axis reports the number of extractable and PA memorized sequences as a function of model size for 4- and 50-token suffixes. The thicker lines denote the proportion of extractable-memorized sequences that are also PA memorized, with their scale indicated on the secondary y-axis.

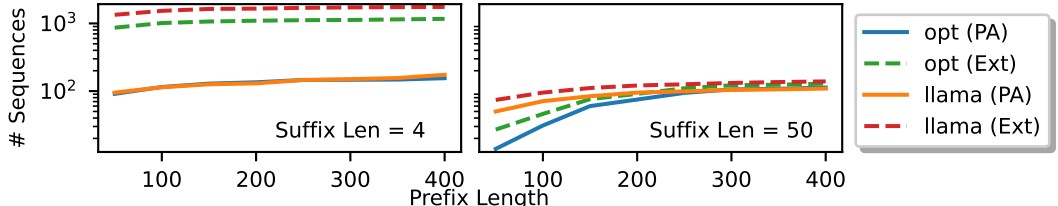

Figure 3: Number of a) Extractable, and b) PA Memorized Sequences as a function of model prefix length for two types of suffixes.

**Decreasing proportion of PA memorized sequences.** Secondly, Figures 2 and 4a show that as model size increases, the proportion of PA memorized sequences generally *decreases*. This suggests that larger models are increasingly able to reproduce verbatim text by generalizing from common or near-duplicate data, rather than through true memorization—a finding consistent with recent work Liu et al. (2025).

## 4.5 EFFECT OF PREFIX LENGTH ON DISCOVERING PA MEMORIZATION

In Figure 3 we plot the number of samples labeled as PA memorized and extractable memorized as a function of the length of prompt $p$ provided to the model. Unsurprisingly, we observe that longer $p$'s can discover more PA memorized sequences. However, note that the PA memorized of 4 token Named Entity suffixes does not benefit as much from longer prefixes as the 50 token suffixes. Again, this is likely due to the fact that many of these Named Entities (*e.g.,*United States of America) are popular occurrences in web-text, and likely do not need specific prompts to be produced correctly.

## 4.6 QUALITATIVE ANALYSIS

Table 2 shows a few sequences with 4-token Named Entity suffixes that have high and low PA memorization scores (i.e., $\frac{P(s|p)}{P(s)}$). We present more such examples in Appendix C. We note that all of the sequences with low scores have suffixes that are places, political figures, etc. We also attempt to verify how "rare" the high-scoring sequences are. While it is infeasible to determine rarity precisely within a dataset with trillions of tokens, we perform a web-search and note that the high-scoring sequences have very few search results. In fact, the highest scoring sequence in Table 2 has just a single search result, which is precisely the sequence that is included in the dataset. We also note that high-scoring suffixes do appear to be more "artifact-like" than their low-scoring counterparts, which appear to be more natural text. This is more apparent in longer sequences, such as the examples presented in Appendix C.

## 4.7 LIMITATIONS

One limitation of PA memorization compared to counterfactual memorization is that $P(s; M)$ can be large due to either many near-duplicates *or* even due to many exact copies of $p\|s$. This is illustrated in Section 3.3, where we attempt to reduce the statistical likelihood of $s$ by replacing near-duplicates with fewer exact copies. However, as shown in Figure 4b, both $P(s; M)$ and $P(s \mid p; M)$ increase

| Score (Low) | Sequence $(p\|s)$ | Score (High) | Sequence $(p\|s)$ |
|---|---|---|---|
| 2.9 | …and is a tributary to **Saginaw Bay** | 4052 | …t1="Sea Zone" t2=" **South Atlantic Sea Zone** |
| 3.0 | …special prosecutor **Leon Jaworski** | 3358 | …misguided members of the **Autonomie Club** |
| 3.7 | …Jack Germond and **Jules Witcover** | 2544 | …I'm watching Gore's **Warmista-Fest** |
| 4.0 | …Hospital had received **Hill-Burton** | 1560 | …PLUS Gold certification.- **Corsair Gold AX850** |

Table 2: Examples of low and high scoring $p\|s$. $s$ in each sequence is in bold text.

when exact copies are added, even though near-duplicates are removed to lower the commonality of $s$. As a result, $\frac{P(s|p;M)}{P(s)}$ increases very *slowly* as more exact copies are added (y-axis in Figure 1), indicating our metric may be less effective than counterfactual memorization at distinguishing whether a high $P(s)$ arises from near-duplicates or from exact copies.

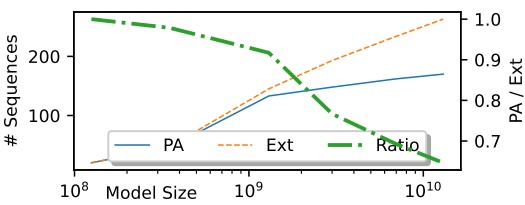 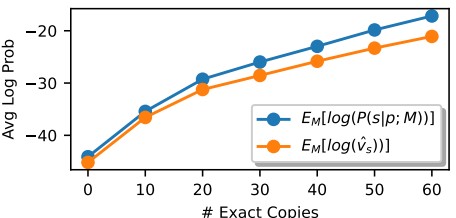

(a) Memorization as a function of model size for 1K sequences from the SATML challenge dataset.

(b) $P(s \mid p)$ and $P(s)$ as a function of the number of exact copies of $p\|s$ in the training dataset.

Figure 4: Results on the SATML Challenge Dataset (a), and $P(s \mid p)$ and $P(s)$ from Section 3.3

## 5 RELATED WORK

Carlini et al. (2019) was one of the first works to examine memorization in LLMs, and did so by introducing canaries during training and measuring their perplexity compared to random data during inference. Carlini et al. (2021; 2022) measured leakage of training data and examined memorization scaling laws in LLMs. Yu et al. (2023) provide new methods to be able to increase leakage of training data from LLMs. Hayes et al. (2024) explores probabilistic metrics for quantifying memorization. Lukas et al. (2023) study leakage of PII in LLMs. Kassem et al. (2024) leverage one LLM for prompt-optimization to leak data from another LLM. Biderman et al. (2024) attempt to predict which sequences will be memorized by an LLM using lower-compute trial runs. Kim et al. (2024) introduce the notion of sequence-likelihood, the metric that we use for our analysis. Tirumala et al. (2022) explore scaling laws and examine memorization dynamics throughout training. Lu et al. (2024) study scaling laws for "fact memorization" within LLMs. Huang et al. (2022) study trends in both memorization, and *association* of PII, where the latter involves leakage through prompts that are not verbatim training sequences. Feldman (2020) introduces the idea of the "long-tail", where they show that models have a tendency to memorize rare, out-of-distribution samples. Maini et al. (2023) explore whether memorization can be localized to certain parts of the model. Ippolito et al. (2023) argue that focusing solely on verbatim memorization provides an incomplete picture of privacy risks. Hartmann et al. (2023) provides a systematic overview of the work done in the memorization space. Morris et al. (2025); Schwarzschild et al. (2024) use compression-based techniques in order to estimate if sequences are memorized. In particular Cohen et al. (2024); Morris et al. (2025) aim to also disentangle memorization from generalization, however, their method relies on having a model that embodies the true distribution of data, and it is unclear if this is practical in the large-scale settings we see today.

## 6 CONCLUSION

In this work, we propose a novel method to identify whether a sequence is counterfactually memorized without having to train different baseline models. Our method shows that at worst 90% of sequences labeled as memorized by prior metrics, are in fact, statistically likely sequences that could be produced without specifically training on the exact sequences. This highlights how traditional metrics may overstate memorization in LLMs by labeling generic sequences as memorized.

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

# A  PROOF FOR THEOREM 2

**Theorem 2**  $Var[\hat{v}_s] = \frac{1}{c} \cdot Var[P(s \mid p_i)] \leq \frac{1}{4c}$

*Proof.*  Since $p_i$ are independent, we can reduce:

$$Var[\hat{v}_s] = Var[\frac{1}{c}\sum_{i=1}^{c} P(s \mid q_i)]$$

$$= \frac{1}{c^2} Var[\sum_{i=1}^{c} P(s \mid q_i)]$$

$$= \frac{1}{c^2} \sum_{i=1}^{c} Var[P(s \mid q_i)]$$

$$= \frac{1}{c^2} \cdot c \cdot Var[P(s \mid q_1)]$$

$$= \frac{1}{c} \cdot Var[P(s \mid p_i)]$$

From Popoviciu's inequality on variances,

$$Var[\hat{v}_s] = \frac{1}{c} \cdot Var[P(s \mid p_i)] \leq \frac{1}{4c}$$

As in Theorem 1, the transition from conditioning on $q_1$ to $p_1$ follows from the equality of their distributions. $\square$

# B  GENERIC SEQUENCES FOR THRESHOLD CALCULATION

In this section, we provide a list of some short and long simple sequences that we use to calculate the threshold $n$ in Definition 2. The short sequences are used to calculate the threshold for the short, 4-token suffixes, while the long sequences are used to calculate the threshold for the longer 50-token suffixes. In each case, each sequence is divided into equal tokens of prefix and suffix used to calculate $\frac{P(s|p)}{P(s)}$. Recall that $n$ is simply an average of $\frac{P(s|p)}{P(s)}$

- Please let me know if you have any questions,

- Thank you for your time and consideration,

- I look forward to hearing from you soon,

- If you have any concerns, please don't hesitate to contact me,

- Don't miss out on this limited-time offer,

- Our mission is to provide the best service,

- Learn more about our features and pricing options,

- Thank you for taking the time to read this message. I am writing to provide an update regarding our project and to ensure that you have all the information needed to move forward. At this stage, we have completed the initial steps and are preparing to begin the next phase. Please review the attached document, which includes a summary of progress, outstanding tasks, and anticipated challenges. If you have any questions or concerns, feel free to reach out at your earliest convenience.,

- This email is intended to confirm the details of our upcoming meeting. The session is scheduled for next week at the agreed time, and we will be covering several important items on the agenda. Please make sure to review the notes shared earlier so that we can make efficient use of our time together. If you are unable to attend, kindly let me know as soon as possible so we can reschedule or provide you with the necessary updates.

- I hope this message finds you well. I wanted to follow up on the status of the work assigned last month and check whether everything is on track for completion by the deadline. If you are facing any obstacles or require additional support, please do not hesitate to reach out. It is important for us to address any potential challenges early so that the overall timeline remains achievable.,

- In this article, we will discuss the basics of this topic and explain why it is important to understand this concept in the broader context of the field. We will begin with a brief introduction, followed by a step-by-step explanation of the main ideas, and finally provide some examples to illustrate how these principles can be applied in practice. Whether you are completely new to this topic or simply looking for a refresher, this guide is designed to help you gain a clear and practical understanding.,

- The purpose of this guide is to help you get started with the software in a straightforward and beginner-friendly way. First, we will walk through the installation process and highlight common issues that may arise during setup. Then we will cover the essential features you need to know to be productive right away. By the end of this guide, you should have a working environment and a solid foundation that will allow you to explore more advanced aspects of the software at your own pace.,

## C  PA Memorization Examples

In this section, we provide examples of $p\|s$ that have statistically likely $s$ and those that do not. Each sequence is preceeded by it's value of $\frac{P(s|p)}{P(s)}$, presented in colored text. The suffix $s$ is presented in bold text. We note that low-scoring sequences have more natural-appearing english text, which is common in the training data for both OPT and Llama models. However, high-scoring sequences tend to have excessive formatting, non-english characters, URLs, are about niche topics, or are boiler-plate text (e.g., terms and conditions from companies) that appear verbatim several times on the web.

### C.1  Low $\frac{P(s|p)}{P(s)}$ sequences

- 1.48 ining the ship after this crossing." "That's not a problem," said Emma without explanation. The purser still didn't look convinced. "Can you read and write?" Emma would like to have told him that **she'd won a scholarship to Oxford, but simply said, "Yes, sir." Without another word, he pulled open a drawer and extracted a long form, passed her a fountain pen and said, "Fill this in." As Emma began**

- 1.48 "pageset": "S53 476 (Tenn. 1973). Our supreme court stated the rationale for this rule: This well-settled rule rests **on a sound foundation. The trial judge and the jury see the witnesses face to face, hear their testimony and observe their demeanor on the stand. Thus the trial judge and jury are**

- 1.55 obvious) mistake in being unacceptably slow getting back into the play after the puck was cleared over his head into the neutral zone. He rather glided through the neutral zone as Moore passed him by to get in position to collect the loose puck. **In a cruel twist, the Caps would get one back with 2:16 left when – finally – someone was able to collect a loose puck in close and do something with it. Brooks Laich lifted a backhand over Halak almost from**

- 1.68 "Pay her off easy, easy," he screamed to the sailors struggling at the wheel, his voice barely audible over the wind. He squinted into the blinding snow in an attempt to time the turn in a smooth, a patch of **waves smaller than the average. The seas came in groups, with every seventh to ninth attaining the most prodigious heights. Sometimes these larger groups merged with a cross sea reared up from the swift tidal currents, or married together to form a rogue**

- 2.05 fluffy, and great). When cooking first sparked my interest in a real way, I decided I needed to master the biscuit. Not coming from a family of biscuit makers, or from a place known for excellent biscuits, I wasn't **even sure what made a great biscuit, if we are being honest. But I was on a quest for perfection.**

- 2.06 The people of Thebes acclaimed Oedipus as their new king. Monster Mash The Sphinx, the Greeks said, had the head of a woman, the body of a lion, the wings of an eagle, and the **tail of a serpent. In fact, this is how the Greeks dreamed up most of their mythical creatures: by mixing and matching body parts from real animals, like a hideous, nightmarish version of Build-A-Bear Workshop. Let's take**

- 2.21 up sinks: cold water Washing machine(s) Laundry sinks: cold water Laundry sinks: hot water Dryers Ironing facilities Food, drink and groceries **Fresh bread available at the camp site Groceries: limited selection Restaurant (with ample choice) Snack bar Takeaway meals Bar Communal barbecue area Freezing for cooling**

## C.2 HIGH $\frac{P(s|p)}{P(s)}$ SEQUENCES

- 10.32 u7684 u6B65 u9AA4u3002 u5FAE u4FE1 u5C0Fu7A0B u5E8F u5BF9 u4E8Eu767B u5F55 u7684 u8BBEu8BA1 uFF0C u66F4 u662Fu7528 u4E4B u4E8E u65E0u5F62 **uFF0C u5728 u6574 u4E2A u7528 u6237 u4F7F u7528 u8FC7 u7A0B u4E2D u90FD u662F u65E0 u611F u77E5 u7684 uFF0C u4E00 u8FDB u5C0F u7A0B u5E8F u5176 u5B9E**

- 7.47 money, now banks are closed..." "Now I don't have cash with me..." "Anto', how much have you got?" "Not a single lira, love." "You see..." "What can we do now?" "Don't **you worry, it does not matter." "We'll do as we have always done." "Here is a million for you!" "It's too much, madam." "Thanks, madam." "No, you must take it." "**

- 6.43 /1332) :star: - Learning Simple Algorithms from Examples. ['pdf'](http://proceedings.mlr.press/v48/zaremba16.pdf) - Learning to Trans **duce with Unbounded Memory. ['arxiv'](https://arxiv.org/pdf/1506.02516.pdf) - Listen, Attend and Spell. ['arxiv'](https://**

- 4.59 be issued to the original card that was used. The refund amount will include only the amount paid by you after any discount or reward was applied to the returned item(s) and it will not include any shipping charge paid by you unless you are returning **a damaged, defective, or the wrong item was sent to you. The wedding dress is perhaps the most carefully chosen dress a woman will ever wear, and an amazing wedding ceremony could be the most unforgettable moment in a woman's life.**

- 3.99 surface and prevent the possibility of sustaining burns unnecessarily while shaving. How can I prevent skin burns? Yes, you can. You have to adopt and adhere to the following best practices to be able to accomplish this particular feat: **Shave in the right Direction: In most instances, this problem arises when the hair is shaved against the direction of the strands thereof. To avoid this problem, it is advisable that you shave in the direction of the hair growth. This is**

- 3.24 ,006 (u5468u5B50) u3061u3087 u3063 u3068 uFF61— u3069 u3046 u3057 u305F u306E uFF1F— u3046 u3093 uFF1F—**,011 u4F1A u793E u306B u884C u304B u306A u304D u3083 uFF61 (u5468 u5B50) u3044 u3044 u304B u3052 u3093 u306B u3057 u306A u3055u3044uFF61 u27A1241 00:**

- 2.78 s costing to fill the car and get groceries! http://no-apologies-round2.blogspot.com/ AmericanborninCanada lol wolfie. Are you a foghorn like I am? http:// **tinyurl.com/wwsotu Wolfie ON my good days! If I went on American Idol, they'd just pay me the top prize to never come back again! However, my wife is a lovely singer and**

- 2.67 , offices, and hotel). The mine-site and township were heavily polluted with asbestos and tailings from the mine were distributed around the town (5). These women have been followed up since 1973 at national and state death and cancer registries as well **as frequent questionnaires to ascertain smoking histories and demographic information. Their mortality and cancer incidence have been reported elsewhere (6-8). The studies have ethics approval from the University of Western Australia Human Research Ethics Committee. A disease cluster is an occurrence**

- 2.65 "parent":"z9iGIu1Pt","text":"", "mid":"z9iZtx1BP","date":"2012-12-11 22:35:08"},{" **kids":[], "uid":"1927311555","parent":"z9iJ1w3Q7", "text":"","mid":"z9iZjcvK8","date":"2012-**

- 2.52 that $N^{\lambda}=0$ and let $n\in\mathbb{Z}_{>0}$ be minimal such that $N^{\lambda-n\alpha}\neq 0$. Let $w\in$ **N^–"lambda -n"alpha ˝\$ be a non-zero element. Using the PBW Theorem, we may write \$w="sum ˙–i=0˝^nc˙if^i"overline –f˝^"–**

- 2.47 , pacing around a long chart unrolled like a hound's tongue across the floor. "We've backed away from the pedigrees in the past year." Spencer's shoe brushes the edge of the pedigree chart. It is **a long genetic octopus, a family tree with arms and legs that tangle and cross, as do those of most degenerate families. Spotted throughout are symbols, keyed on the side. A dark black circle signifies Insane. A hollow**

## D    LLM USAGE

LLMs were used sparingly to polish a few sentences for better grammar and clarity.

