# OpenReview forum: "Efficient Metric for Distinguishing Memorization from Generalization in Large Language Models"
_ICLR.cc/2026/Conference — Submitted to ICLR 2026_

### Official Review · Reviewer_VzFo · 2025-10-25

**Soundness:** 1
**Presentation:** 2
**Contribution:** 2
**Rating:** 0
**Confidence:** 4

**Summary:**

Counterfactual memorization is proposed as a measure for memorization that can distinguish from generalization. Counterfactual memorization, however, requires retraining the model for each sample, with and without including it in training, resulting in an expensive and often impractical computational requirement.

The authors propose prior-aware memorization, which does not rely on training, yet can simulate memorization that is correlated with counterfactual memorization. The goal is to filter out statistically likely suffix. The authors propose that if a suffix incurs high generation probability even with the model is prompted with *any* randomly sampled text, then the suffix *may* be generated due to generalization, and hence not memorization.

There are results relating/contrasting with existing implementation of counterfactual memorization. Also, the authors filter out a large amount of sequences, that would be classified as memorization in earlier models to be a case of generalization.

**Strengths:**

The motivation of improving the computation inefficiency of counterfactual memorization is indeed justified.


The connection between conditional probability to Bayes formula is interesting: the conditional probability of generating a suffix $Pr(s | p)$ given a prefix is equal to $\Pr(s) * \frac{\Pr(p | s)}{\Pr(p)}$, where the first term denotes statistical frequency of the suffix, and the second term is the relative belief. The rest of the paper is based on using this decomposition to filter out statistically likely suffixes. I like this intuition, however the main results in Section 3 and onwards is flawed as explained below.

**Weaknesses:**

- Equation 1 motivates to decompose the conditional probability into suffix probability and belief ratio. However, definition 2 (and later analysis) **does not consider** belief ratio: $\frac{\Pr(p | s)}{\Pr(p)} \ne \frac{\Pr(s | p)}{\Pr(s)}$. If the goal is to approximate $\Pr(s | p)$, I do not understand why computing $\frac{\Pr(s | p)}{\Pr(s)}$ is relevant. This is fundamentally **wrong**. I encourage the authors to clarify this in rebuttal -- I am willing to increase my score if I am mistaken.

Other comments.
-  Line 37: John Doe may be statistically common, but it does not mean that asking the LLM the prefix *the murder was committed by*, it would generate John Doe. This does not make sense. Also, if some suffix is statistically common in *training*, it does not necessarily become generalization out of context. Reference Ghosh et al., as cited in the paper, has proposed a better differentiation between memorization and generalization.
- Line 102-104. Better explanation is required on the belief ratio. One should not naively assume $\Pr(p)$ is small relative to $\Pr(p|s)$.
- Line 186: what does the author mean by matching the distribution of $q_i$ and $p_i$? It is not immediately clear.
- Line 194 is not mathematically grounded.
- References are poorly formatted throughout the paper. There is no parenthesis before and after any citation.

**Questions:**

Address the weakness.

---

> ### Author Response · Authors · 2025-11-17
>
> Thank you for the constructive comments on the work!  We address below the weaknesses outlined in your review.
>
> - **Equation 1**.  Perhaps our explanation was too sparse and should be made clearer.  It ***is*** the case that $\frac{P(p|s)}{P(p)} = \frac{P(s|p)}{P(s)}$, and this follows from Bayes' theorem which states that:
>
>   $P(A \mid B) = \frac{P(B \mid A) \cdot P(A)}{P(B)}$
>
>   Dividing both sides by the prior probability, $P(A)$, yields:
>   $\frac{P(A \mid B)}{P(A)} = \frac{P(B \mid A)}{P(B)}$
>
>   In our context, with $A=p$ (prefix) and $B=s$ (suffix), the equality is:
>   $\frac{P(p \mid s)}{P(p)} = \frac{P(s \mid p)}{P(s)}$
>
>   In other words, the two ratios are therefore equivalent, motivating the decomposition in Equation 1.
> - **Approximating $P(s \mid p)$.**  Note that we are **not** approximating $P(s|p)$, but rather computing it empirically using the LLM's next-token prediction mechanism (see Section 3.1).  What we are approximating is $P(s)$, the prior probability of the suffix, and we are doing this using the unbiased Monte-Carlo estimator from Section 3.2.
>
> - **Relevance of $\frac{P(s \mid p)}{P(s)}$.** Once we have computed $P(s)$ and $P(s \mid p)$, we can compute the ratio $\frac{P(s \mid p)}{P(s)}$, which, per the above, is equal to $\frac{P(p \mid s)}{P(p)}$.  We will clarify this point in the manuscript.
>
> - **Line 37.** Perhaps there is some confusion about our approach.  We are simply stating that a statistically popular suffix is *more likely* to be generated (from the same prompt) than an unpopular one.  In other words, the fact that suffix occurs with ``high probability'' may simply be an artifact of this observation.
>
> - **Ghosh et al.**  This work ultimately suffers from the same drawback as counterfactual memorization [1], in that computing contextual memorization requires training a new baseline model (i.e., without the target sequences) for \emph{every} sequences whose memorization is to be measured.  This approach is computationally infeasible on  large, pretrained models, which is the impetus for our approach.  We will further clarify this in the text.
>
> - **Lines 102-104.**  Please note that the text states that ***if*** $P(p)$ is small relative to $P(p \mid s)$ ***then*** the relative belief ratio is large.  No claim is made for other cases.
>
> - **Line 186.**  This is simply a restatement of the definition condition that ``samples $q_0, q_1, \ldots q_{c-1}$ chosen independently and identically according to the distribution of prefixes $p_i \sim V^*$.  We apologize for the confusion.
>
> - **Line 194.**  Perhaps the confusion here is with respect to the terminology $p_i \sim V*$ in Definition 3?  This $i$ is not significant, and we will clarify this in the updated version by rewriting $p \sim V*$ instead.  The statement of line 194 itself is a direct consequence of one of the [standard properties](https://www.colorado.edu/amath/sites/default/files/attached-files/definetti.pdf) of independent and identically distributed (iid) random variables --- that if $q_i$ and $q_1$ are iid, then $E[g(q_i)] = E[g(q_1)]$ for any measurable function $g(\cdot)$ (in this case, $g(x) = P(s \mid x)$).
>
> - **Formatting References.** Thank you for pointing this out; we will add parentheses around all citations in the revised manuscript.
>
> - Citations:
> [1] C. Zhang, D. Ippolito, K. Lee, M. Jagielski, F. Tram`er, and N. Carlini.
> Counterfactual memorization in neural language models. Advances in Neural
> Information Processing Systems, 36:39321–39362, 2023

---

> > ### Comment · Reviewer_VzFo · 2025-11-26
> >
> > Thanks for the clarification. I get the trick using Bayes' formula, which should have been made clearer in the paper.
> >
> > Still, I cannot comprehend what $\frac{\Pr(s|p)}{\Pr(s)}$ captures. The model is not trained to learn $\Pr(s|p)$, which is to generate suffix given prefix. What is the fundamental thinking here?

---

> > > ### Author Response · Authors · 2025-12-02
> > >
> > > - **$P(s\mid p)$**. The model ***is*** trained to learn $P(s\mid p)$, as per the standard cross-entropy objective function applied to the Language Modeling Objective (i.e., the next-token prediction task [1,2] as further elaborated [here](https://wandb.ai/madhana/Language-Models/reports/Language-Modeling-A-Beginner-s-Guide---VmlldzozMzk3NjI3#:~:text=ReferencesRecommended%20Reads-,What%20Is%20Language%20Modeling?,that%20has%20the%20highest%20probability.) and [here](https://marinafuster.medium.com/cross-entropy-loss-for-next-token-prediction-83c684fa26d5).
> > >
> > > - **$\frac{P(s\mid p)}{P(s)}$**. The intuition behind the ''Relative Belief Ratio'' (see [this article](https://www.sciencedirect.com/science/article/pii/S2001037015000549)) is described on line $102$ of the text and further elaborated in the explanation of Definition $2$.
> > > Intuitively, we want to label a sequence $p \Vert s$ as memorized when:
> > >   - $P(s\mid p)$ is high, meaning that the model predicts $s$ with high likelihood when prompted with its prefix $p$.  This would happen if the model strongly learned this during training, since, as discussed above, the model is trying to maximize $P(s\mid p)$ for every prefix-suffix pair during  training on next-token prediction), and
> > >   - The unconditional probability $P(s)$ is small (i.e., we do not want to count a sequence $p \Vert s$ as memorized if $s$ is very statistically likely in the model's training data).
> > >
> > > If these two conditions are met, then the more concise formulation $\frac{P(s\mid p)}{P(s)}$ is also high (although the converse might not hold).
> > >
> > > Citations:
> > >
> > > [1] Y. Bengio, R. Ducharme, P. Vincent, and C. Jauvin. A neural probabilistic
> > > language model. Journal of machine learning research, 3(Feb):1137–1155,
> > > 2003.
> > >
> > > [2] D. Jurafsky and J. H. Martin. Speech and Language Processing. Pearson,
> > > 2024. Draft of 3rd edition, Chapter 3: N-gram Language Models Chapter
> > > 10: Transformers.

---

### Official Review · Reviewer_Yckr · 2025-10-25

**Soundness:** 3
**Presentation:** 2
**Contribution:** 2
**Rating:** 6
**Confidence:** 3

**Summary:**

This paper introduces Prior-Aware (PA) Memorization, a computationally efficient metric designed to distinguish between true memorization and statistical generalization in LLMs. State-of-the-art approaches like counterfactual memorization require retraining models with and without specific data points, making them infeasible for modern LLMs. The authors propose a substitute measure that can be computed directly from a pre-trained model’s outputs without retraining. They observe that 55–90% of sequences previously classified as memorized were generalizable. Larger models show a decline in PA-memorized sequences, implying they rely more on generalization. Additionally, PA scores are positively correlated with Counterfactual Memorization, confirming the metric’s validity.

**Strengths:**

1. The authors propose a practical metric for memorization analysis that can be used with any model without retraining.
2. The derivation of PA-Memorization from Bayes' rule is interesting.
3. The methodological section is well-written.
3. Empirical results suggest that the metric is useful.

**Weaknesses:**

1. The experimental results need to be polished. The setup is clear but Sections 4.4 and 4.5 are not very clear in their takeaways.
2.  The abstract mentions "..55 − 90% of the sequences that would be classified as “memorized” in earlier models are, in fact, generalizable sequences." The paper doesn't seem to discuss on this aspect in the main paper.
3. More thorough experiments could have been undertaken -- for e.g. how is PA-memorization going to be for rare but sensitive items?
4. Figure visualisation can be improved -- e.g. for Figure 2, suffix length could have been a title or in the caption. Please consider writing the model names in the case as they are -- OPT, Llama (instead of opt, llama).
5. Spelling mistakes -- line 87 : "extrctable' --"extractable" , line 153 : "beleif" -- "belief".

**Questions:**

Same as Weaknesses above and below
1. What is meant by opt(Ratio) or llama(Ratio) -- Is it the fraction of sequences memorized as per relative belief ratio? It's not mentioned.
2. How can it be inferred that with decrease in PA-memorized sequences, "larger models are increasingly able to reproduce verbatim text by generalizing from common or near-duplicate data, rather than through true memorization" ? An experiment on showing this effect of generalization could have been more convincing.

---

> ### Author Response · Authors · 2025-12-02
>
> Thank you so much for your feedback! Our responses are below:
> - **Polishing Text/Figures.** Thank you for your note. We will polish these results
>
> - **"..55 - 90% of the sequences"** These values come directly from OPT(Ratio) and LLama(Ratio). These ratios in those diagrams refer to the number of sequences we label as memorized over the number of sequences extractable memorization (EM) labels as memorized for each model type. We will clarify this in the text.
>
> - **Generalization vs. Memorization.** Our observation that "larger models are increasingly able to reproduce verbatim text by generalizing from common or near-duplicate data, rather than through true memorization" is supported by the trends in OPT(Ratio) and LLama(Ratio). Specifically, the ratio of sequences identified by our metric (PA memorization) to the total number of verbatim sequences (Extractable Memorization, EM) generally decreases as model size increases.This indicates that while larger models generate more verbatim text, a growing proportion of this output consists of statistically likely sequences, which our PA metric correctly filters out. Consequently, this generation is driven not by rote memorization of specific training instances ($p \Vert s$), but by the model’s ability to generalize to high-frequency patterns ($s$).

---

### Official Review · Reviewer_8jkg · 2025-10-28

**Soundness:** 2
**Presentation:** 2
**Contribution:** 2
**Rating:** 2
**Confidence:** 3

**Summary:**

This paper examines the problem of separating extractable memorization that arises out of generalization -- those being generally statistically likely -- from those that are memorized because a certain sequence was present in the pretraining corpus. The existing definition of this -- counterfactual memorization -- is highly inefficient to measure because it relies on an ability to retrain models for which a particular sequence was not present in the corpus. This work introduces prior aware (PA) memorization as an efficient approach to correct measurements of memorization for the presence of statistically likely common sequence. They estimate the general statistical frequency of a suffix by measuring its probability under randomly sampled pre-training prefixes. Using their score, they identify that 55-90% of memorized sequences can be attributed to generalization.

**Strengths:**

This paper explores the important problem of more accurately memorizing the amount that large language models memorize. They identify an important problem facing existing definitions -- statistically common text sequences -- and devise an intuitive approach for adjusting memorization measurements to account for them.

**Weaknesses:**

(1) I am a bit unsure of the novelty of the problem formulation of this work. For example, in Carlini et.al (2021), there is already an extensive discussion of "trivially" memorized subsequences which are those that are common and generally have high likelihood. In fact, they already propose ways to "baseline" whether a text is statistically common using independent models, zLib entropy, and perturbation such as casing. Thus, I am not sure that the observation/problem statement is particularly novel; nor is it made clear at all why the methodology introduced in this paper is superior to the existing works proposed in Carlini et.al.

(2) As the paper notes, the sequences used to test memorization are primarily randomly sampled from various public sources. As a result it seems somewhat unsurprising that a majority of such sequences are those that are more statistically likely under the model. Thus, I am not sure how easy it is to interpret the claim "We discover that for the largest models, 55-90% of the sequences that would be classified as ``memorized'' in earlier models are, in fact, generalizable sequences."  (as this fraction would depend significantly on the underlying source of sequences used to probe memorization).

(3) Similarly, I also feel dubious about the claim that "Decreasing proportion of PA memorized sequences. Secondly, Figures 2 and 4a show that as model size increases, the proportion of PA memorized sequences generally decreases. This suggests that larger models are increasingly able to reproduce verbatim text by generalizing from common or near-duplicate data, rather than through true memorization". This may very well be the case for the types of sequences used to probe memorization in this paper. However,  this does not rule out that larger models would also memorize the rarer data at higher rates (which may not be significantly represented in the experimental settings here to be accurately measured).

**Questions:**

Please respond to the points raised in weaknesses.

---

> ### Author Response · Authors · 2025-11-20
>
> Thank you for your detailed and thoughtful comments.  Our responses follow:
>
> - **Novelty.** The novelty of the paper lies in its contribution of a metric that a) has theoretical guarantees on error, b) is easier to compute than counterfactual memorization, and c) has been tested to correlate with counterfactual memorization.  We will clarify this in the revised version of the work.
> While we agree that the work Carlini et al (2021) does discuss ''trivially'' memorized sequences, as you note, identifying each sequence requires some baseline that either involves an alternate model or some other approximation of a baseline model (e.g., zlib entropy). Unfortunately, it is very difficult to argue that these baseline models are representative of the correct training setup (e.g., trained in the exact same way on the same data minus the target sequence), and thus these approaches are prone to providing false positives on memorization (see the SATML position paper by Zhang et al (2024).
> As an example, Zhang et al. notes that it is easy to have a baseline model ''not perform well'' on a target sequence not because the target sequence was absent from the training set, but because the baseline training setup is sufficiently different from the actual model's training setup (e.g., if the baseline model is much smaller, or has been trained for less time, etc.). The most rigorous way to measure memorization is thus through counterfactual memorization, which is cost prohibitive to measure in realistic settings.
> We must also stress that all of the aforementioned baselines are empirical in nature, and consequently have no guarantees on whether the scores of the baseline models actually correlate with the natural experimental setup of having a separate baseline per target sequence. We, on the other hand, not only have guarantees on how our metric performs, but we also validate its correlation with counterfactual memorization.
>
> - **Interpretation of Results.** We apologize for the confusion, this sentence should be better worded as: ''our results show that 55-90%  classified as memorized in earlier ***methods*** ...''. i.e., methods, not models. This statistic compares our method with existing methods of measuring memorization on the same set of sequences to argue that existing methods overestimate memorization due to their inability to differentiate statistically likely sequences from genuine memorization.
>
> - **Decreasing proportion of PA memorization**. We agree with your point about the proportion of sequences that are memorized in larger models. Our intent was to ensure that the sequences used for the memorization analysis were representative of the training data's actual distribution, so as to best understand the model's behavior in that context.  In so doing, we remain consistent with prior quantification attempts such as Carlini et al (2022)
>
> - **Citations**
> - N. Carlini, D. Ippolito, M. Jagielski, K. Lee, F. Tramer, and C. Zhang.
> Quantifying memorization across neural language models. In The Eleventh
> International Conference on Learning Representations, 2022.
> - J. Zhang, D. Das, G. Kamath, and F. Tram`er. Membership inference at-
> tacks cannot prove that a model was trained on your data. arXiv preprint
> arXiv:2409.19798, 2024.

---

### Official Review · Reviewer_3GmT · 2025-10-28

**Soundness:** 1
**Presentation:** 2
**Contribution:** 1
**Rating:** 2
**Confidence:** 4

**Summary:**

Given an output (or suffix $s$) and input (or prefix $p$) during training of a neural language model (NLM), the authors tackle the problem of identifying whether the output was memorized by the NLM. They do this while trying to differentiate between statistically common (suffixes that are presented a large number of times in the training set) and truly memorized suffixes.
In order to do this, they introduce the prior-aware memorization metric (PAm) based on the prior $\frac{\mathbb P(s|p)}{\mathbb P(s)}$. PAm is briefly compared with another memorization metric, also addressing this dichotomy (counterfactual memorization), in Section 3.3.3. PAm is also compared with the classical extractable memorization (Em), which does not differentiate between the two types of memorization, to show how PAm can solve the main problem.

**Strengths:**

S1. The problem of addressing the memorization of non-statistically likely responses is well-motivated and an important issue for NLMs.

S2. The motivation for the use of a different metric besides extractable memorization (Em) --- Since it can be large due to $s$ being a statistically likely output rather than a truly memorized sequence--- and counterfactual memorization (Cm) --- Due to its computational intractability --- was well presented.

S3. The metric presented, prior-aware memorization (PAm), was well motivated by the gaps presented by S1 and S2, as well as by the explanation in Section 2.

**Weaknesses:**

W1. The rigorousness of the mathematical notation could be vastly improved. For instance, without knowing the domain of the sequences $p\|s$, one has to infer how the conditional probabilities are defined. Furthermore, multiple mathematical objects used throughout the paper were not introduced ($E,\hat =,\overset{\triangle}{=},\dots$) or sometimes used differently [$m$ (lines 148 vs 153) and $n$ (lines 148 vs 150)]. Furthermore, the arguments of the metrics, as introduced in (6) and (7), are also unclear --- Figure 1 seems to utilize a set of sequences ($p\|s$) while Def. 2 and (5) use a single pair $(p,s)$ as arguments.

W2. The work presents Cm [Zhang 2023], which addressed the problems of Em. PAm was introduced as a computationally cheaper alternative to Cm. However, the main evaluation does not consider this newer and better-suited metric.

W3. The presentation of the paper could be greatly improved. For instance, the Figures contain unnecessary plots (the ratios of two very clearly observable quantities) and are unordered (Figures 4a and 2 are presented before Figures 3 and 4b). Further, the wording is sometimes confusing (for example, "*We evaluate these metrics on various sizes 2 recent generative models* ($\cdots$)", line 320).

W4. The evaluation of PAm does not provide enough evidence to support the main claim of being an alternative that does not require the retraining of models. This is mainly driven by the lack of a comparison with Cm, but also by other evaluation decisions (See questions Q1, Q3, and Q4).

W5. Some influential metrics addressing the same problems of Cm and Em are missing in the related work and in the experiments (see Wang 2025). Without a proper analysis of related metrics (both theoretically and experimentally), there is not enough novelty in the article.

**Questions:**

In order to improve the manuscript, I kindly ask the Authors to assess the following questions and suggestions.

Q1. Lines 91-107 introduce the main idea for PAm. While it is true that $\mathbb P(s|p)$ does not differentiate between statistically likely suffixes and believable suffixes for the prefix, I do not see how using $\frac{\mathbb P(p|s)}{\mathbb P(p)} = \frac{\mathbb P(s|p)}{\mathbb P(s)}$ (the prior used for PAm) can differentiate between the two. As $\mathbb P(s|p) = \frac{\mathbb P(s,p)}{\mathbb P(p)}$, an increase on the joint probability $\mathbb P(s,p)$ does not need to be smaller than the increase of the individual probabilities of $p$ and $s$. This will, in turn, make $\frac{\mathbb P(s|p)}{\mathbb P(s)}$ bigger for a statistically likely sequence. This might call into question the validity of the prior (and hence, of PAm) to detect such differences. Since the near-duplicates used in the experiments are very dissimilar semantically (see Table 1 in the article ), an in-depth study of the distribution of PAm in a setting using more similar near-duplicates is needed to support the claim.

Q2. As mentioned before, Cm is inversely correlated with duplication of the pairs $(p,s)$ (see Section 5 in Zhang 2023). How can the authors explain the increase in Cm in Figure 1? Further, the experiments in Section 3.3.3 are not explained in detail, and conflict with the definition of Cm and PAm given in Def. 2 and (5) --- See W1. Could the authors clarify this?

Q3. Cm is a well known and popular metric to address memorization on NLMs. Yet, it was barely analyzed in this article. Furthermore, PAm is presented as a computationally simpler alternative to Cm. To be able to be presented as an alternative, a proper comparison of its performance with Cm needs to be presented. Without this comparison, the evidence does not support the claim. Could the authors consider an in-depth comparison with this metric?

Q4. Why do the authors center in the use of a threshold value, rather than on the study of the distribution of the memorization metrics vs interesting statistics of the sequences, like in Zhang 2023, Wang 2025, and Jiang 2024? Understanding how the metric behaves for the entire distribution of values, not only the tails, is also relevant (and also eliminates the need to tune the thresholds).

Q5. Some influential papers also tackling the same problem seem to be missing. More importantly, Wang 2025 also proposes an alternative to Cm that differentiates between generalization and memorization, while being computationally cheaper. Were the authors aware of this article?


While the authors propose an interesting and much simpler alternative to Cm, they do not compare PAm with Cm. Furthermore, a more modern alternative to Cm [Wang 2025] where not included. This alternative tackles the problems of Cm, while being computationally cheaper (the same claims as PAm). Without a comparison between all of them, there is **(a)** not enough evidence to support the main claims and **(b)** not enough novelty. Upon tackling Q1-Q5 above, I will reconsider my evaluation of the manuscript.


**References**:

[Zhang 2023] Zhang, C., Ippolito, D., Lee, K., Jagielski, M., Tramer, F., & Carlini, N. (2023). Counterfactual Memorization in Neural Language Models. Advances in Neural Information Processing Systems, 39321–39362.

[Jiang 2024 ] Jiang, M., Liu, K. Z., Zhong, M., Schaeffer, R., Ouyang, S., Han, J., & Koyejo, S. (2024). Investigating Data Contamination for Pre-training Language Models. https://arxiv.org/abs/2401.06059

[Wang 2025] Wang, X., Antoniades, A., Elazar, Y., Amayuelas, A., Albalak, A., Zhang, K., & Wang, W. Y. (2025). Generalization v.s. Memorization: Tracing Language Models' Capabilities Back to Pretraining Data. The Thirteenth International Conference on Learning Representations.

---

> ### Author Response · Authors · 2025-12-02
>
> Thank you so much for your feedback! We respond to your comments below, and break down our responses into two parts due to character limits.
>
> - **W1:**
>     - **Operators**: Thank you for your note on mathematical operators. $E$ refers to Expectation, and $\triangleq$ in Equation $5$ refers to it's definition in the original counterfactual memorization paper [1], from where this definition was cited. $\Vert$ refers to the concatenation operator, so $p \Vert s$ refers to a concatenation between strings $p$ and $s$. We will explicitly define these terms in the paper. The ''$\mid$'' operator refers to the conditional probability operator, not the ''such that'' operator in the set builder notation. Finally, $\hat{=}$ refers to ''is estimated by''.
>
>    - **Line 148 vs 153:** $m$ and $n$ were accidentally switched in the explanation of Lines 150-154. We will correct this in the revised draft.
>
> - **W2, W4 and Q3: Comparison between PAm and Cm:** We ***do*** provide a comparison between Cm and PAm in Section $3.3$, and a larger scale evaluation of PAm in Section $4$. Perhaps there is some confusion on what Section $3.3$ actually measures, and we try to clarify this below; please see ``**Q2: Section 3.3.3**''
>
> - **Q1: Interpretation of the PAm Ratio.** We agree with the reviewer's derivation that our metric is equivalent to $\frac{P(s,p)}{P(s)P(p)}$. Consequently, if a suffix $s$ is statistically likely (large $P(s)$), the overall ratio decreases. This behavior is by design. We intend for PAm to be small for statistically likely sequences; this allows our threshold $n$ to effectively filter out sequences $p \Vert s$ where the generation is driven by the popularity of $s$ rather than the context $p$.
>
>     Furthermore, we can decompose the joint probability $P(s,p)$ into $P(p|s)P(s)$. In the ratio, the $P(s)$ terms cancel out, leaving $\frac{P(p|s)}{P(p)}$. Since $\frac{P(p|s)}{P(p)} = \frac{P(s|p)}{P(s)}$, the mathematical equivalence holds: a larger $P(s)$ inherently drives the ratio down.
>
>     In conclusion, regardless of the algebraic form—whether defined via joint or conditional probability—the metric consistently yields lower values for statistically likely suffixes. This aligns perfectly with our argument that such sequences should not be classified as "memorized."
>
> - **Q2: Clarification on Section 3.3.3 (Definition vs Measurement).** There may be a misunderstanding regarding how Cm and PAm are operationalized, and we will try to clarify this in the revision. While Definitions 2 and 4 provide the general theoretical descriptions, Equations 6 and 7 specify how these quantities are computed in practice. And so, Figure $1$ and all of Section $3.3$ does actually evaluate Cm against PAm. Cm (Definition 4) conceptually relies on the performance difference between a target and a baseline model. Equation 6 implements this by measuring the difference in log probability. Specifically, $E_M$ represents the expected performance on $p \Vert s$ across models trained on that sequence, while $E_{M'}$ represents the expected performance of baseline models (those whose training sets do not contain $p \Vert s$). Similarly, Equation 7 is the direct computational translation of Definition 2 (PAm). It measures the ratio $\frac{P(s \mid p)}{P(s)}$ in log space to quantify the dependency between the prefix and the suffix.
>
> - **Q2: exact-duplicates vs near-duplicates: Clarification on Cm behavior (Exact vs. Near-duplicates).** The concern regarding Cm may stem from confusion over the distinction between exact copies and near-duplicates. We align with Section 5 of Zhang et al. [1], which demonstrates that when the number of ***near***-duplicates is large, Cm is low. This relationship is explicitly depicted in Figure 1. As detailed in Lines 250-265, the x-axis progresses from a high prevalence of near-duplicates (left) to a low prevalence (right). To support this, each data point in Figure 1 includes a tuple indicating the precise counts of exact copies and near-duplicates. We will update the caption and text to ensure this distinction is immediately clear to the reader.
>
> Citations:
>
> [1] Zhang, C., Ippolito, D., Lee, K., Jagielski, M., Tramer, F., & Carlini, N. (2023). Counterfactual Memorization in Neural Language Models. Advances in Neural Information Processing Systems, 39321–39362.
>
> [2] Wang, X., Antoniades, A., Elazar, Y., Amayuelas, A., Albalak, A., Zhang, K., & Wang, W. Y. (2025). Generalization v.s. Memorization: Tracing Language Models' Capabilities Back to Pretraining Data. The Thirteenth International Conference on Learning Representations.

---

> > ### Author Response · Authors · 2025-12-02
> >
> > - **W5 & Q5: Comparison with Wang et al. [2].** We acknowledge that the definition in Wang et al. [2] shares conceptual similarities with our work and counterfactual memorization. However, their approach faces significant computational hurdles in large-scale settings. Specifically, while their method avoids training separate baseline models, it necessitates computing a task-gram model, which requires iterating over the entire dataset to calculate frequency counts for target sequences.In contrast, the core contribution of our metric is its efficiency. We eliminate the need for both separate baseline models (required by counterfactual memorization) and full dataset iteration (required by Wang et al.). Instead, we rely on estimating $P(s)$ via model inference using a set of randomly sampled prefixes (see Section 3.2). This makes our approach significantly more scalable and efficient to compute. We will clarify this difference in the text.
> >
> > - **Q4: Evaluation Methodology.** Our primary objective was to re-evaluate the scaling laws of memorization [3] using our more selective metric. We utilized a thresholding approach because it facilitates the clear analysis of trends against independent variables, such as model size. However, we agree that analyzing the underlying distributions provides additional insight. We are happy to include additional results detailing the distribution for specific model sizes and prefix lengths in the final revision.
> >
> > - **W3.** Thank you for your note; we will polish the figures in the refined draft.
> >
> > Citations:
> >
> > [1] Zhang, C., Ippolito, D., Lee, K., Jagielski, M., Tramer, F., & Carlini, N. (2023). Counterfactual Memorization in Neural Language Models. Advances in Neural Information Processing Systems, 39321–39362.
> >
> > [2] Wang, X., Antoniades, A., Elazar, Y., Amayuelas, A., Albalak, A., Zhang, K., & Wang, W. Y. (2025). Generalization v.s. Memorization: Tracing Language Models' Capabilities Back to Pretraining Data. The Thirteenth International Conference on Learning Representations.
> >
> > [3] N. Carlini, D. Ippolito, M. Jagielski, K. Lee, F. Tramer, and C. Zhang. Quantifying memorization across neural language models. In The Eleventh International Conference on Learning Representations, 2022.

---

### Official Review · Reviewer_SbhA · 2025-11-01

**Soundness:** 1
**Presentation:** 2
**Contribution:** 2
**Rating:** 2
**Confidence:** 5

**Summary:**

This work proposes a computationally effective metric, i.e., prior-aware memorization, to approximate the counterfactual memorization. The authors have first conducted correlation analysis of the proposed metric and the counterfactual memorization measure on 350 GPT-2 models trained with injected sequences that are either exact duplications or near-deduped. The authors then use the proposed metric to measure memorization on pre-trained Llama and OPT models, which suggests that 55-90% of data that was classified as memorized were generalizable sequences.

**Strengths:**

The paper aims to solve an interesting problem, i.e., the counterfactual-based definition of memorization is accurate, however, computationally expensive to measure.

**Weaknesses:**

* The key assumption made in the paper (line 53-54) seems not well justified.
   * The authors argue that the differences between "genuine memorization" and "statistical likelihood" can be measured by "testing whether a sequence retains high probability even when the model is prompted with randomly sampled text from the training data. If a suffix appears with high confidence across many unrelated prefixes, its likelihood arises from generalization". This might be true if a memorized sequence happens to be split between the prefix $t$ and suffix $s$ for testing. However, if $s$ by itself corresponds to the memorized sequence, such as digits of $\pi$, i.e., $s=3.1415926535897932$, will it be counted as sequences with high "statistical likelihood"? $s$ might have low perplexity due to being repetitively optimized in training, especially for long sequences of $s$, where the model learns that reconstruction $s$ only depends on the first few tokens of $s$. This contradicts the counterfactual memorization definition since removing all occurrences of $\pi$ from training data, it is highly unlikely that a model can generalize this sequence from others due to its irregularity.
   * Although this assumption is empirically verified in section 3.3 through a correlation analysis on synthetic training runs, the setup here seems quite different from pre-training setup, where the near-dedup sequences are mostly noisy non-natural language sequences, while in actual pre-trianing data, near-dedup usually come in the form of textual variations and templated texts. Moreover, the comment on line 246-246 is inaccurate "Thus, injecting highly overlapping sequences may not realistically reflect modern training regimes, where such examples are typically removed." In fact, sequences in pre-training corpus can still repeat anywhere from 100 to tens of thousands of times, depending on the dedup algorithm used. These differences in the setup make it hard to tell if the assumption still holds on pre-training data distribution.

* **False claims on the property of SATML Challenge dataset**
   *  The authors claim that the SATML Challenge, **"consists of 1-eidetic sequences (sequences where each p||s is known to occur only once in The Pile)" (L343-L344), which is factually incorrect**. This dataset comes from https://github.com/google-research/lm-extraction-benchmark. The official document clearly states the properties of the sequences as "easy-to-extract", i.e., "there exists a prefix length that causes the model to generate the suffix string exactly" and "well specified", i.e., "given the 50-token prefix, there is only one continuation such that the entire sequence is contained in the training dataset." Neither of these descriptions indicate these sequences are "1-eidetic " or "occur only once in The Pile". Quite the opposite, these sequences actually occur very frequently in training data, which is why they are easy to extract (one could easily verify with tools like infini-gram, for example, the first sequence in the training split occurs about 7084 times in the Pile).
   * This discrepancy might explain the "surprising" findings that the authors have reported on L375-377, where the authors claim that "around 40% of sequences are common in nature". Considering the fact these sequences occur extremely frequently in the Pile, it perhaps not just these sequences are common in nature, but rather, there are just multiple exact copies. Overall, this might weaken the findings on line 21, where authors claim there are more "generalizable sequences" than measures used in previous work.

* No comparison with other memorization measurements that do not require training: For example, [Schwarzschild et al. 2024](https://proceedings.neurips.cc/paper_files/paper/2024/file/66453d578afae006252d2ea090e151c9-Paper-Conference.pdf) proposed a compression-based metric that is also motivated by the computation cost of counterfactual memorization. How does the metric proposed in this work compares with theirs in terms of efficiency and accuracy?

* Related work not discussed:
     * Contractual memorization: [Huang et al. 2024](https://aclanthology.org/2024.emnlp-main.598.pdf), [Lesci et al. 2024](https://aclanthology.org/2024.acl-long.834.pdf); both work uses counterfactual memorization to study properties of memorization in LLMs
     * Relationship between memorization and generalization: [Prashanth et al. 2025](https://openreview.net/pdf?id=3E8YNv1HjU)

**Questions:**

* Memorization and generalization are not well-defined in the paper.
   * On line 21, what does "generalizable sequences" mean here? Generalizable in what sense?
   * On line 53, what does "genuine memorization" mean here? What are the quantitive criteria?
   * On line 402, the authors claim that "larger models are increasingly able to reproduce verbatim text by generalizing from common or near-duplicate data, rather than through true memorization"; What does "true memorization" mean here? The authors cited Liu et al. 2025 as evidence, could the authors kindly point out which observations from Liu et al. 2025 support this argument?

* Could the authors elaborate the experiment setup in section 3.3. In particular, (1) what is the injected sequence length (2) are most near dedup sequences replaced with random tokens, i.e., the injected near-dedup sequences are no longer in natural language?

---

> ### Author Response · Authors · 2025-11-20
>
> Thank you for your detailed and thoughtful comments.  Our responses follow, and are broken down into two parts due to character limits:
>
> - **Lines 53 - 54.**  Please note that our definition (Definition 2) does not refer to memorized sequences on their own, but only with respect to a prefix and suffix. This follows from related work such as [1, 3]. If the suffix $s_\pi = 3.1415926535897932$ occurs often but almost only in the context of a prefix (e.g., $p_\pi = \text{``$\pi$  is equal to''}$) only then do we say that $p_\pi || s_\pi$ is memorized.  Alternatively, if the pair is $p_1 = 3.1415926$ and $s_1 = 535897932$, then $p_1 || s_1$ will be considered memorized for the same reason, as $p_1 \Vert s_1$ recur often in the text, and $s_1$ does not occur frequently (or at all) with other prefixes.
> Note also that we do not claim that we will catch ***all*** memorized sequences, just that Prior-Aware memorization implies memorization.  We will make this more clear in our revised submission.
>
> - **Synthetic training runs** We acknowledge that the training runs for our smaller scale experiments have synthetic sequences. As such, we will augment Section 3.3 with sequences naturally occur in the training dataset. However, here too, our initial results indicate that if a suffix is statistically likely (e.g., $p_2$ = ''I live in the '' and $s_2$ = ''United States of America''), $s_2$ can be easily generated with other prefixes, indicating that $p_2 \Vert s_2$ is not actually memorized.
>
> - **SATML dataset.** Perhaps the confusion here is that we are not categorizing the Pile dataset, which, as you note is not 1-eidetic, but rather the prefix-suffix pairs in the challenge dataset, which have the property that any full prefix corresponds to one and only one full suffix.  Thus, each sequence $p_i \Vert s_i$ in the challenge dataset ***is*** 1-eidetic, as described in the original definition from Carlini et al (2021). This is because, as you correctly stated, "given the $50$-token prefix, there is only one continuation such that the entire sequence is contained in the training dataset''.  We will clarify this confusion in the revised text.
> In this context, our argument still holds --- even though each $p_i \Vert s_i$ occurs only once in the Pile, these sequences are ''common'' in nature and thus $s_i$ can be easily generated even without prompting with the original $p_i$.
>
> - **Schwarzschild et al. 2024.** We do not provide a head-to-head comparison with Schwarzschild et al. (2024) because their definition of memorization is fundamentally different from ours, and this complicates any direct comparison of results.  Schwarzschild et al. (2024) argue that a suffix is memorized if it can be generated by a highly compressed prefix. For instance, consider the prefix $p_1$ = ''The murder was committed by'' and the suffix $s_1$ = ''John Doe.'' Their metric would label the generation of $s_1$ as memorized since there are short prompts that are likely to generate this simple suffix. In contrast, our definition of memorization is derived from Carlini et al. (2022). Intuitively, we argue that the sequence $p_1 \Vert s_1$ is labeled as memorized only if the suffix $s_1$ is generated with high likelihood specifically from the prefix $p_1$ (found in the training data), and not from an arbitrary prefix. This stricter criterion allows us to effectively handle statistically likely or common sequences, such as the aforementioned ''John Doe.''
>
> - **Prior Work** We were not aware of the two articles provided and will include an updated comparison in the revised draft.

---

> ### Author Response · Authors · 2025-11-20
>
> - **Terminology.** Thank you for highlighting the need for clearer terminology; we will ensure these terms are tightened in the revised draft. When we use the term "generalizable" in the context of a sequence $p_i \Vert s_i$, we are referring to the suffix $s_i$ being statistically probable and therefore having a high likelihood of generation not just with its specific training prefix $p_i$, but also with other, distinct prefixes that may occur in the input space. Such an output is considered an instance of reconstructive inference or non-rote generation. Conversely, a sequence is considered "genuinely memorized" if its suffix $s_i$ is statistically unique and can only be generated with the specific prefix $p_i$ with which it appeared in the training data. This represents the kind of rote memorization we seek to isolate and measure.
>
> - **Liu et al. (2024) - The Role of Generalization in Verbatim Generation** The findings in Liu et al. directly support our hypothesis regarding the overestimation of memorization in existing metrics. This study investigates the ability of LLMs to generate "verbatim" sequences that were absent from their training data but were present in a near-duplicate form (e.g., variations in punctuation or capitalization).
>   - **Key Insight**: The models exhibit a robust generalization capacity to synthesize the exact target text from these near-duplicate instances, suggesting the output is a result of reconstructive inference rather than rote memorization.
>   - **Scaling Effect**: The paper confirms that this generalization ability scales with model size (Table 1), providing intuitive evidence that models acquire advanced capabilities to extrapolate from similar inputs.
>   - **Implication for Memorization Metrics**: This phenomenon suggests that existing memorization metrics, which typically flag any reproduction of a training sequence $p_i \Vert s_i$ as "memorized," are conflating true memorization with outputs that could have been generalized from related-but-not-identical data. Our proposed metric addresses this by incorporating a more nuanced definition that filters out statistically common or reconstructible suffixes. Crucially, since larger models show both increased generalization from near-duplicates and higher rates of reported extractable memorization (defined in Line 87), it becomes fundamentally ambiguous whether the observed increase in 'memorization' across scales is due to genuine rote memorization or merely an enhanced ability to generalize and reconstruct near-duplicate content.
>
> - **Experimental Details.** Injected sequences were $100$ tokens, where $50$ were prefix tokens, and $50$ were suffix tokens. Yes, the injected sequences were replaced with random tokens. Post submission, we have repeated the experiments with natural text, and will provide those results in our revised draft.
>
>
>
>  **Citations**:
>
> [1] N. Carlini, D. Ippolito, M. Jagielski, K. Lee, F. Tramer, and C. Zhang.
> Quantifying memorization across neural language models. In The Eleventh
> International Conference on Learning Representations, 2022.
>
> [2] N. Carlini, F. Tramer, E. Wallace, M. Jagielski, A. Herbert-Voss, K. Lee,
> A. Roberts, T. Brown, D. Song, U. Erlingsson, et al. Extracting training data
> from large language models. In 30th USENIX security symposium (USENIX
> Security 21), pages 2633–2650, 2021.
>
> [3] J. Hayes, M. Swanberg, H. Chaudhari, I. Yona, I. Shumailov, M. Nasr,
> C. A. Choquette-Choo, K. Lee, and A. F. Cooper. Measuring memorization
> in language models via probabilistic extraction. In Proceedings of the 2025
> Conference of the Nations of the Americas Chapter of the Association for
> Computational Linguistics: Human Language Technologies (Volume 1: Long
> Papers), pages 9266–9291, 2025

---

### Meta-Review · Area_Chair_pPgH · 2026-01-06

**Summary:**

This paper proposes Prior-Aware Memorization (PAm), a computationally inexpensive metric intended to distinguish rote memorization from generalization-driven verbatim generation in large language models. The paper tackles an important and timely question: how to separate memorization from generalization in verbatim generation settings, where counterfactual memorization is computationally prohibitive at modern scale.

The reviews are mixed but overall lean negative. Multiple reviewers question the core assumption that high probability of a suffix across randomly sampled prefixes reliably indicates generalization rather than memorization in real-world corpora, where duplication, templating, and natural near-duplicates are common. The current evaluation relies heavily on synthetic injected-sequence experiments, including random-token near-duplicates, which may not faithfully reflect realistic pretraining data distributions. At the same time, although the method is positioned as an efficient alternative to counterfactual memorization, the empirical comparison is not sufficiently direct or comprehensive: reviewers requested clearer head-to-head comparisons with counterfactual memorization, and experimental comparisons with other non-retraining memorization proxies. These concerns are compounded by clarity and notation issues. While the rebuttal acknowledges several of these points and promises clarifications and additional experiments, they are not yet reflected in the current submission.

In summary, the core idea is promising, and the rebuttal helps clarify some conceptual aspects. However, the submission does not yet establish robustness under realistic pretraining data characteristics nor provide sufficiently comprehensive empirical comparisons against strong existing alternatives. Combined with presentation and notation issues and ambiguities around key claims and datasets, the paper is not yet at the level required for acceptance. On balance, I recommend rejection at this time.

**Reviewer Concerns:**

The rebuttal clarified several points, including the derivation in Equation 1 and aspects of the interpretation of the results.

**Reviewer Scores:**

Reviewer VzFo indicated that they may slightly increase their score, as the rebuttal clarified Equation 1 and the associated motivation.

---

### Decision · Program_Chairs · 2026-01-26

Reject